DATA RELEASE

# Snail intermediate host occurrence recorded by citizen scientists in rural Uganda and the Democratic Republic of the Congo

Noelia Valderrama-Bhraunxs[1,2,*], Larissa Bonifacio[1], Julius Tumusiime[3], Germain Kapour[4], Daisy Namirembe[3], Casim Umba-Tolo[3], Grace Kagoro-Rugunda[3], Patrick Mitashi-Mulopo[4], Joule Mandinga[5,6], Liesbet Jacobs[7] and Tine Huyse[1,*]

1 Department of Biology, Royal Museum for Central Africa, Tervuren, Belgium
2 Department of Earth and Environmental Sciences, KU Leuven, Leuven, Belgium
3 Department of Biology, Faculty of Science, Mbarara University of Science and Technology, Uganda
4 Department of Tropical Medicine, University of Kinshasa, Kinshasa, Democratic Republic of the Congo
5 National Institute of Biomedical Research, Kinshasa, Democratic Republic of Congo
6 Biomedical Department, University of Kikwit, Kikwit, Democratic Republic of the Congo
7 Institute for Biodiversity and Ecosystem Dynamics, University of Amsterdam, Amsterdam, The Netherlands

## ABSTRACT

Snail-borne parasitic diseases, such as schistosomiasis and fascioliasis, pose significant public health and economic challenges worldwide. Schistosomiasis affects over 250 million people globally, with most cases in sub-Saharan Africa, while fascioliasis contributes substantially to livestock morbidity and economic losses. Freshwater snails (*Biomphalaria*, *Bulinus*, and *Radix* spp.) act as intermediate hosts, making their surveillance critical for disease control. Mass drug administration alone is insufficient, as high reinfection rates highlight the need for complementary strategies, including targeted snail control. To address limited malacological capacity and logistical constraints, the ATRAP project trained 50 citizen scientists in Uganda and the Democratic Republic of the Congo to monitor intermediate host snails at the genus level. Between 2020 and 2023, citizens recorded 31,490 snail occurrences. Data quality was ensured through automatic validation and manual verification of submitted snail pictures. This rigorously curated dataset, combining citizen science with expert validation, provides valuable insights for mapping snail distributions, identifying high-risk transmission areas, and developing sustainable, cost-effective snail control strategies.

**Subjects** Ecology, Biodiversity, Taxonomy

**Submitted:** 08 April 2025

\* Corresponding authors. E-mail: noelia.valderrama@africamuseum.be; tine.huyse@africamuseum.be

Preprint submitted at https://africarxiv.ubuntunet.net/handle/123456789/2004

Included in the series: *Vectors of human disease* (https://doi.org/10.46471/GIGABYTE_SERIES_0002)

## BACKGROUND

Snail-borne parasitic diseases present major public health and veterinary challenges across tropical and subtropical regions [1]. Schistosomiasis, caused by parasitic flatworms of the genus *Schistosoma*, affects more than 250 million people globally, with the majority of cases occurring in sub-Saharan Africa (SSA) [2]. Chronic infections are associated with anaemia, stunted growth, cognitive impairment, and reduced aerobic capacity [3]. Depending on the organs affected, schistosomiasis can also lead to hematuria, infertility, liver and urinary tract fibrosis, and even bladder cancer [3]. In livestock, schistosomiasis is caused by

*Schistosoma bovis*, *Schistosoma curassoni*, and *Schistosoma matthei*, and is widespread, contributing significantly to animal morbidity [4]. Fascioliasis, a parasitic disease caused by the trematodes *Fasciola hepatica* and *Fasciola gigantica*, poses a dual threat to human health and livestock productivity. It is estimated to affect between 2.4 and 17 million people globally, with the highest burden in Africa and Asia [5]. In livestock, fascioliasis causes substantial economic losses, estimated in billions of dollars each year, due to reduced productivity [6].

Freshwater snails of the genera *Biomphalaria* and *Bulinus* act as intermediate hosts for schistosome parasites, while *Lymnaeidae* snails from the *Radix* genus are the primary intermediate host for *Fasciola* liver flukes [7–9]. *Biomphalaria* spp. transmits *Schistosoma mansoni*, the causative agent of intestinal and hepatic schistosomiasis, whereas *Bulinus* spp. transmits both *Schistosoma haematobium* and *Schistosoma intercalatum*, which cause urinary and intestinal schistosomiasis, respectively [7, 8].

Despite the widespread implementation of mass drug administration (MDA) programs in high-burden areas, reinfection with schistosomiasis remains common, emphasizing the need for integrated control measures [10, 11]. In Uganda, for instance, *S. mansoni* remains highly endemic, particularly in communities around the Great Lakes region, where prevalence rates continue to exceed 50% despite the implementation of regular MDA campaigns [12, 13]. Recognizing the limitations of drug-based interventions alone, the World Health Organization (WHO) has acknowledged that controlling intermediate snail hosts, which are responsible for the transmission of the disease, is essential for sustainable disease control. As part of its 2030 roadmap for schistosomiasis elimination, the WHO now recommends integrating MDA with snail control and improvements in the water, sanitation and hygiene (WASH) infrastructure [11].

Snail control interventions have shown promise in reducing transmission rates [10, 14]. However, effective and targeted snail control requires detailed mapping of snail habitats and transmission hotspots, a process constrained by the shortage of trained malacologists and the logistical difficulties of monitoring large, remote areas [15]. While chemical molluscicides such as niclosamide are effective in reducing *Bulinus* spp. and *Biomphalaria* spp. populations, concerns about environmental safety and sustainability persist [14, 16]. Therefore, enhancing snail surveillance and habitat mapping is crucial for informing targeted, ecologically responsible control efforts [17].

Across SSA, schistosomiasis is heterogeneous, but Uganda and the Democratic Republic of the Congo (DRC) present two high-priority settings. In Uganda, intestinal schistosomiasis caused by *S. mansoni* is endemic in more than 65% of the total districts (146) and can exceed 50% prevalence in lakeshore communities. In comparison, urogenital schistosomiasis caused by *S. haematobium* is limited to less than 3% of the districts [13]. The DRC reports the third-highest number of cases of schistosomiasis in SSA [18, 19], both *S. mansoni* and *S. haematobium* are endemic nationwide, with poor sanitation and limited access to healthcare contributing to high rates of reinfection [19, 20]. However, data on snail distribution remain scarce, creating a significant challenge for effective snail control measures [21].

To address these challenges, we launched the Action Towards Reducing Aquatic Snail-borne Parasitic Diseases (ATRAP) initiative in 2019, a citizen science (CS) project that engages local communities in monitoring snail intermediate hosts to identify potential disease transmission hotspots.

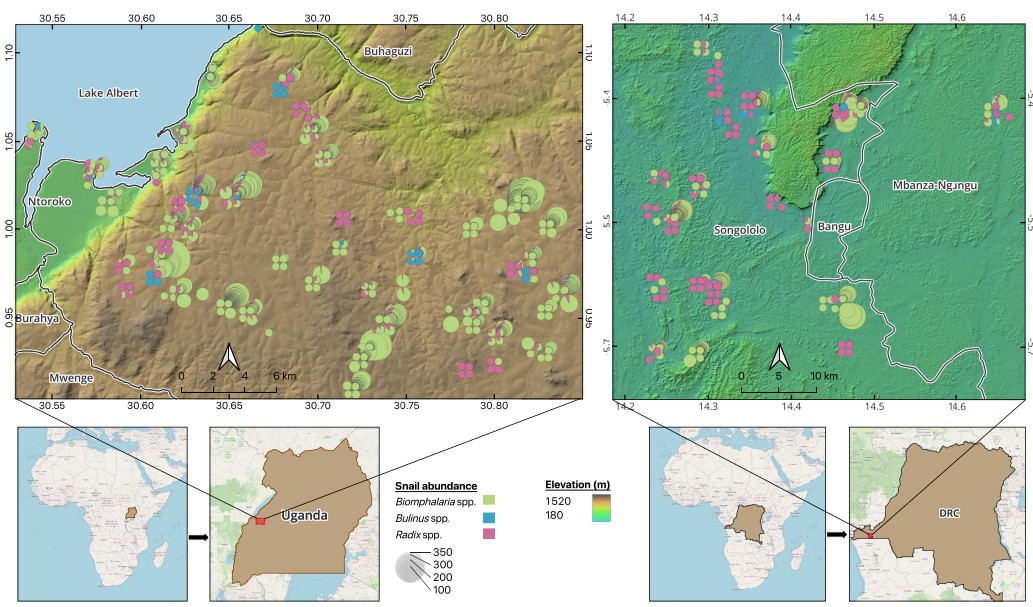

**Figure 1.** Snail intermediate host abundance based on CS reports submitted between 2020 and 2023 in (a) Uganda and (b) the DRC. Digital Elevation Model from Copernicus GLO-30 [27]. Legends are standard for both study areas.

## Context

The ATRAP project was established to fill critical data gaps in rural areas of Uganda and the DRC, where schistosomiasis prevalence remains high. This CS-based initiative aimed to monitor snails as intermediate hosts and generate fine-scale distribution data to guide targeted control interventions.

While CS projects with structured data collection protocols can enhance the scope of monitoring, produce high-quality actionable data, and increase local community engagement [22–24], concerns about data reliability persist due to its non-expert contributors [25]. To address this, ATRAP recruited and trained 25 citizen scientists in each country who carried out weekly snail monitoring from 2020 to 2023; the resulting data were subsequently subjected to rigorous validation protocols to ensure data quality. The project was a collaboration between the Royal Museum for Central Africa in Belgium, Mbarara University of Science and Technology in Uganda, and the University of Kinshasa in the DRC.

## METHODS

### Study area

#### *Ugandan citizen science network*

Data collection for the Ugandan CS network took place in the Lake Albert region (Figure 1a), an area characterized by significant altitudinal variation, ranging from 620 m in the lowlands to 1,400 m in the upland areas shaped by the Great Rift Valley. The lower-lying regions near the lake are remote and difficult to reach by road, presenting logistical challenges for fieldwork. The sampling sites captured a variety of hydrological conditions and were classified into four main types: springs, streams, wetlands, and lake shores. A detailed description of these site types can be found in [24] and GigaDB [26].

### *DRC citizen science network*

In the DRC, the CS network operated in the Kimpese region of Kongo Central province (Figure 1b). In contrast to the varied topography of the Ugandan study area, this region is relatively homogeneous, characterized by plateaus and rolling hills at elevations ranging from approximately 250 to 400 m. The Lukunga River and its tributaries form an extensive hydrological network, serving as essential water sources for irrigation during the rainy season [28]. Most of the sampling locations corresponded to streams derived from the Lukunga River.

## Sampling design and protocol

In both study areas, sampling locations were identified through a preliminary survey, prioritizing sites where community members frequently engaged in water-related activities such as swimming, washing, bathing, water collection, and open defecation/urination practices, which increase the risk of snail-borne disease transmission. The study included locations where snail intermediate hosts were present, as well as areas where they were absent during initial field assessments.

Citizen scientists conducted standardized snail sampling and submitted reports through KoboCollect, documenting freshwater snails that serve as intermediate hosts for *Schistosoma* and *Fasciola* parasites. The survey protocols used in each country are available in GigaDB [26]. In Uganda, where site types were more heterogeneous, a sampling protocol with a 10 m radius was recommended at each site. In contrast, in the DRC, citizen scientists collected snails 5 m upstream and downstream of the contact point. For streams less than 3 m wide and 50 cm deep, sampling was conducted on both sides of the stream. Each survey lasted approximately 30 minutes, during which the citizen scientists used a hand-held scoop net with a 2 m metallic handle while wearing protective gear (latex gloves and gumboots) [22]. Snails were identified and sorted based on morphological characteristics learned during training, with counts and photographs recorded for each specimen. Reliable species-level separation within the genera *Biomphalaria*, *Bulinus*, and *Radix* generally requires a combination of shell morphology, soft tissue anatomy, and molecular analysis [29–31]; therefore, citizens recorded identifications only to the genus level. Snails were categorized into the genera *Biomphalaria*, *Bulinus*, and *Radix*, with a 'pool' category reserved for all other taxa. All findings were uploaded to a central server for researcher access and verification [22].

Notably, all 12 *Biomphalaria* species reported from SSA are competent intermediate hosts of *S. mansoni* [32]. In our Ugandan study area, monthly expert surveys with molecular confirmation identified four *Bulinus* species: *B. forskalii*, *B. globosus*, *B. nasutus productus*, and *B. truncatus* [33, 34]. Although *B. globosus* is the recognized vector of *S. haematobium* in Uganda, infected snails and clinical cases of urogenital schistosomiasis have not been reported in the areas surveyed. However, in this part of Uganda, *B. nasutus* acts as the intermediate host of *S. bovis*, and infected snails have been reported [33]. In our DRC sites, only *B. globosus* and *B. truncatus* have been reported, and both are proven intermediate hosts of urinary human schistosomiasis [21]. In addition, Brown's book [32] highlights a single widespread species, *Lymnaea natalensis* (currently *Radix natalensis*), as the principal intermediate host of *Fasciola* spp. in Africa. Although the ATRAP CS project records snails only to the genus level, the data are still epidemiologically informative: *Biomphalaria* spp. detections flag possible areas of transmission of intestinal schistosomiasis in both countries,



**Table 1.** Filtering for ATRAP datasets.

| Description | Uganda | DRC |
|---|---|---|
| Original dataset | 6,570 | 6,891 |
| Excl. no sampling | 5,611 | 6,891* |
| Excl. dates outside range | 5,607 | 6,862 |
| Date diff. > 7 days ('today' vs. 'end') | 5,560 | 6,514 |
| Excl. inaccurate GPS readings** | 5,018 | 6,438 |
| Reports with sampling time between 1 and 60 min | 4,868 | 5,813 |
| Excl. reports where all 3 snails records were invalid | 4,863 | 5,801 |

The three snail genera evaluated were *Biomphalaria*, *Bulinus*, and *Radix*.
*Non-sampling instances were not recorded in the DRC.
**Accuracy greater than 5 m.

*Bulinus* spp. detections indicate urogenital risk in the DRC and mainly bovine transmission in the Ugandan sites, and *Radix* spp. detections mark potential hotspots for human livestock fascioliasis throughout the study region.

## Data validation and quality control

A robust validation protocol was implemented to minimize reporting errors, which are defined as discrepancies between the data collected by citizen scientists and the information submitted in the reports [22, 24]. The protocol included the use of skip logic, prioritized multiple-choice questions, and employed a semi-automatic validation system to detect invalid entries. Additionally, citizen scientists received continuous feedback and participated in annual refresher training sessions to improve the accuracy of their data [22, 24]. For further details on the semi-automatic validation process, see [22].

In addition, manual validation was performed by reviewing the attached field photographs to verify genus identification and flag any possible misidentification. Each snail genus (*Biomphalaria*, *Bulinus*, and *Radix*) was independently verified to ensure that errors in one genus did not affect the validity of other records. The validated reports were then saved to a CSV file for subsequent data cleaning and analysis. This validation protocol was applied to the ATRAP datasets for Uganda and the DRC, which included 6,570 and 6,891 CS reports, respectively.

### Sampling events and associated occurrences

In this paper, a report is defined as a sampling event in which a citizen scientist collects data at a specific location and time. Each sampling event can contain up to three associated occurrences corresponding to the abundance of *Biomphalaria*, *Bulinus*, or *Radix* snails.

### Reports exclusions

In Uganda, 959 reports were excluded due to participants' inability to sample, citing reasons such as weather, illness, or lack of motivation. Additional exclusions included reports with dates outside the sampling period (4), 'end' dates more than 7 days apart from the form's 'today' date to ensure week-bound sampling (47), inaccurate Global Positioning System (GPS) readings (542), and sampling durations outside the 1–60 minute range (150). Misidentification of the three snail genera (*Biomphalaria*, *Bulinus*, and *Radix*) led to the exclusion of 5 reports. After all exclusions, 4,863 valid reports remained for analysis (Table 1).

In the DRC, reports were excluded for similar reasons, including dates outside the sampling period (29), 'end' dates more than 7 days apart from the form's 'today' date (348), inaccurate GPS readings (76), and sampling durations outside the 1–60 minute range (625). Reports containing invalid data for all three snail genera (12) were excluded (see section below). After exclusions, 5,801 valid reports were retained for analysis (Table 1).

### *Invalid associated occurrences and identification errors*

The approach to handling identification errors and invalid associated occurrences differed between Uganda and the DRC due to logistical constraints.

In Uganda, validation was conducted in parallel with data collection, allowing real-time feedback to citizen scientists. This immediate verification process significantly reduced errors and improved data quality. Among the 4,863 valid reports from Uganda, a total of 14,589 associated snail occurrences were recorded. Of these, only 206 occurrences (1.41%) were identified as errors, either due to misidentifications or invalid associated occurrences. Specifically, 32 occurrences of *Biomphalaria*, 97 of *Bulinus*, and 77 of *Radix* snails were excluded from the analysis due to their low frequency, ensuring minimal impact on the overall dataset integrity.

In contrast, data validation in the DRC was conducted after data collection, as real-time feedback was not possible. To address this limitation, the original validation script was refined to correct misidentified occurrences rather than excluding them. Identification errors were systematically corrected, affecting 448 occurrences of *Biomphalaria*, 765 of *Bulinus*, and 2,921 of *Radix* snails. The high number of corrections for *Radix* spp. was largely attributed to frequent misidentifications involving *Melanoides* spp., a snail genus with a right-facing shell opening similar to *Radix* snails. This morphological similarity likely contributed to misclassification by citizen scientists in the DRC.

Following the correction of identification errors, invalid associated occurrences were handled separately. These were defined as records lacking images or containing images unrelated to snails. A total of 121 occurrences of *Biomphalaria* spp., 46 of *Bulinus* spp., and 129 of *Radix* spp. were excluded from the DRC dataset due to these issues.

### REUSE POTENTIAL

When combined with other data sources, CS snail intermediate host data can play a key role in creating risk maps and establishing vulnerability assessments for the transmission of schistosomiasis and fascioliasis. By identifying areas of high probability of snail occurrence and understanding seasonal variations, these data enhance the precision of targeted interventions. Such insights can guide the prioritization of areas for WASH initiatives and targeted snail-control efforts, improving the efficiency of public health responses. Moreover, vulnerability assessments based on CS data can help identify populations at higher risk, allowing for more targeted interventions, such as MDA, in the most affected areas. By using CS-collected data, health agencies can allocate resources more efficiently, ensuring a more effective approach to reducing the burden of schistosomiasis. Notably, previous studies [24, 35] have demonstrated a high presence/absence agreement between observed and modeled expert and CS data for the Uganda CS dataset, further validating the reliability of CS data for epidemiological applications.

Beyond direct epidemiological applications, the release of snail images offers further research opportunities. These images can support the development of identification



methods [36] and facilitate studies of size distribution within intermediate host snail genera. Integrating image-based analysis with on-site surveillance data further improve fine-scale, cost-effective monitoring contributing to disease control strategies.

## DATA AVAILABILITY

The dataset described here is available on the GBIF-RMCA repository [37]. The associated images (snail photographs) in the RMCA BioCase repository [38, 39]. Bulk downloads of images and other data are available from GigaDB [26].

## EDITORS' NOTE

This paper is part of a series of Data Release articles working with GBIF and supported by TDR, the Special Programme for Research and Training in Tropical Diseases hosted at the World Health Organization, in order to publish datasets on vectors of human diseases [40].

## LIST OF ABBREVIATIONS

ATRAP: Action Towards Reducing Aquatic Snail-borne Parasitic Diseases; CS: citizen science; DRC: Democratic Republic of the Congo; GPS: Global Positioning System; MDA: mass drug administration; SSA: sub-Saharan Africa; WASH: water, sanitation, and hygiene; WHO: World Health Organization.

## DECLARATIONS

### Ethical approval

This study was approved by the Ethics Review Committee of Mbarara Technology with reference number MUREC 1/7 and by the Uganda National Council of Science and Technology with reference number NS148ES for Uganda. For the DRC, by the Department of Tropical Medicine from the UNIKIN with reference number SP-2020-023-MED TROP. In addition, written informed consent was obtained from the citizen scientists who participated in this study for both countries. The extent of disclosure of the information provided by citizen scientists and their facilitation were fully discussed and formalized in a memorandum of understanding before the commencement of the data collection process. The memorandum of understanding was renewed annually after a refresher training to allow citizen scientists to share their experiences and expectations.

### Consent for publication

Not applicable.

### Competing interests

The authors declare that they have no competing interests.

### Author's contributions

NVB: Writing – original draft, Writing – review and editing, Data validation and curation, feedback CS network, Funding acquisition. LB: Writing – original draft, Data validation and curation. JT: surveillance, coordination of the CS network. GK: surveillance, coordination of the CS network. DN: surveillance, coordination of the CS network. CUT: Coordination, Supervision, Project administration, Funding acquisition. GRK: Supervision, Project

administration, Funding acquisition. PMP: Supervision, Project administration, Funding acquisition. JM: Coordination, Supervision, Project administration, Funding acquisition. LJ: Conceptualization, Supervision, Validation, Funding acquisition, Writing – review and editing. TH: Conceptualization, Funding acquisition, Supervision, Writing – review and editing.

## Funding
This research was financed by the ATRAP project of the Development Cooperation program of the Royal Museum for Central Africa with the support of the Belgian Directorate General Development Cooperation and Humanitarian Aid (Grants XM-DAC-2-10-3852, XM-DAC-2-10-3853). NVB is a fellow of the Research Foundation Flanders (FWO) (Fellowships 11L3223N, 11L3225N). The funders had no role in study design, data collection and analysis, decision to publish, or manuscript preparation.

## Acknowledgements
We would like to sincerely thank all the citizen scientists whose contributions made this study possible. In Uganda: Alinda Hassan, Chotum Friday, Atanasi Marisel, Opio Isingoma, Ategeka Augustine, Bahemuka Bob, Masereka Haruna, Sebakara Fobius, Nyamahunge Imelda, Businge Zabron, Tumusiime Janet, Nsenga David, Night Marygoret, Kyaligonza Noeline, Mwesige Robert, Barisigara Gard, Tusingwire Henry, Natabi Specioza, Nakingi Rose, Bamuturaki Charles, Kamukama Josias, Nuwagaba Emmanuel, Tweheyo Julius, Bahungirehe Crezestom, Ategeka Rogers, Unimu Hadijah, and Fuwarinyo Richard. In the Democratic Republic of the Congo: Bulezi Jephirin, Dianzenza Héritier, Diatuke Dinayame, Kiangebeni Mbuta, Dikizekiyo Paul, Kalumpuniko Siluafundiswa, Kavungu Zolan- donga, Kinsengwa Lukwikilu, Landu Lusevakueno, Landu Tazi, Lubanzadio Adèle, Lufuma Lwakanda, Luntadila Dayina, Lusevakweno N'sanda, Luvuma Swamina, Luzolo Matufwene, Mansweki Makuntima, Mantantu Muanga, Mayawila, Mabanza Nlandu, Mena Ntonzi, Ndongala Gracia, Nginamawu Luvevola, Nkumbu Ngina, Nsilulu Zanzambi, Nzinga Emmanuel, Puabu Lusimana, Vantour Kinuani, and Zoladio Sienge.

Without their dedicated efforts, this research would not have been possible. We greatly appreciate their involvement and support throughout the study.

We would also like to express our gratitude to Franck Theeten and Larissa Smirnova for their invaluable assistance in making this data available through the Integrated Publishing Toolkit of the Royal Museum for Central Africa.

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
