## [Reviewer Report]

Indicate in the comments box below whether you are happy with the changes made or if the manuscript is unacceptable.Comments on revised manuscriptAdvice 1: Sampling Protocol Specificity: The manuscript mentions survey protocols in Supplementary Material S1 (not found in the RVT system), which should clearly explain how the "10-meter radius sampling protocol" in Uganda and the "5 meters upstream and downstream of the contact point" in DRC were implemented. Could it be included in the GigaDB? Advice 2: The image data can be archived in the GigaDB.Indicate in the comments box below whether you are happy with the changes made or if the manuscript is unacceptable.Comments on revised manuscriptAdvice 1: Sampling Protocol Specificity: The manuscript mentions survey protocols in Supplementary Material S1 (not found in the RVT system), which should clearly explain how the "10-meter radius sampling protocol" in Uganda and the "5 meters upstream and downstream of the contact point" in DRC were implemented. Could it be included in the GigaDB? Advice 2: The image data can be archived in the GigaDB.

---

## [Editor Report]

Editor’s AssessmentThis Data Release submitted to the GigaByte Vectors of Human Disease series presents data from freshwater snails that are intermediates of hosts of Snail-borne parasitic diseases such as schistosomiasis and fascioliasis that are acute and chronic diseases in many parts of the world, including Africa. The ATRAP Citizen Science project was established to fill critical data gaps in rural areas of Uganda and the Democratic Republic of Congo, where schistosomiasis prevalence remains high. The project being a collaboration between the Royal Museum for Central Africa in Belgium, Mbarara University of Science and Technology in Uganda, and the University of Kinshasa in the DRC. To help scale surveillance and targeted snail control, ATRAP trained 50 citizen scientists to monitor snails as intermediate hosts and generate fine-scale distribution data to guide targeted control interventions. Between the years 2020 and 2023 recording 31,490 snail occurrences that have been added into the GBIF biodiversity repository. Data auditing a peer review helping fix any discrepancies so this is of the highest quality. This data can hopefully play a key role in creating risk maps and establishing vulnerability assessments for the transmission of schistosomiasis and fascioliasis, and the associated imaging data can also support the development of AI-based identification methods.Editor’s AssessmentThis Data Release submitted to the GigaByte Vectors of Human Disease series presents data from freshwater snails that are intermediates of hosts of Snail-borne parasitic diseases such as schistosomiasis and fascioliasis that are acute and chronic diseases in many parts of the world, including Africa. The ATRAP Citizen Science project was established to fill critical data gaps in rural areas of Uganda and the Democratic Republic of Congo, where schistosomiasis prevalence remains high. The project being a collaboration between the Royal Museum for Central Africa in Belgium, Mbarara University of Science and Technology in Uganda, and the University of Kinshasa in the DRC. To help scale surveillance and targeted snail control, ATRAP trained 50 citizen scientists to monitor snails as intermediate hosts and generate fine-scale distribution data to guide targeted control interventions. Between the years 2020 and 2023 recording 31,490 snail occurrences that have been added into the GBIF biodiversity repository. Data auditing a peer review helping fix any discrepancies so this is of the highest quality. This data can hopefully play a key role in creating risk maps and establishing vulnerability assessments for the transmission of schistosomiasis and fascioliasis, and the associated imaging data can also support the development of AI-based identification methods.

---

## [Reviewer Report]

Reviewer name and names of any other individual's who aided in reviewer Yannan FanDo you understand and agree to our policy of having open and named reviews, and having your review included with the published papers. (If no, please inform the editor that you cannot review this manuscript.)YesIs the language of sufficient quality?YesPlease add additional comments on language quality to clarify if needed
Are all data available and do they match the descriptions in the paper? YesAdditional CommentsThe image files should be archived in the BioImage platform https://www.ebi.ac.uk/bioimage-archive/ with an appropriate license. The DOI should be updated in the revised MS. https://biocase.africamuseum.be/images/kobo_congo/ 
https://biocase.africamuseum.be/images/kobo_uganda/Are the data and metadata consistent with relevant minimum information or reporting standards? See GigaDB checklists for examples <a href="http://gigadb.org/site/guide" target="_blank">http://gigadb.org/site/guide</a>YesAdditional CommentsThe dataset is in IPT but does not have the DOI from GBIF. The author needs to finish the GBIF registration and update the GBIF DOI in the MS. - e.g. https://doi.org/10.15468/vs8rf8 Semi-automatic and then manual validation was carried out on this dataset 28,977 citizen science occurrences. Then semi-automatic and then manual validation -> 6570 (UG) & 6891 (DRC) reports. Then exclusions applied (table 1) -> 4910 (UG) & 4921 (DRC) reports.Is the data acquisition clear, complete and methodologically sound?YesAdditional CommentsIs there sufficient detail in the methods and data-processing steps to allow reproduction?YesAdditional CommentsIs there sufficient data validation and statistical analyses of data quality? Not my area of expertiseAdditional CommentsIs the validation suitable for this type of data?YesAdditional CommentsIs there sufficient information for others to reuse this dataset or integrate it with other data?YesAdditional CommentsAny Additional Overall Comments to the AuthorRecommendationMinor Revision

---

## [Reviewer Report]

Reviewer name and names of any other individual's who aided in reviewer Suzete Rodrigues GomesDo you understand and agree to our policy of having open and named reviews, and having your review included with the published papers. (If no, please inform the editor that you cannot review this manuscript.)YesIs the language of sufficient quality?YesPlease add additional comments on language quality to clarify if needed
Are all data available and do they match the descriptions in the paper? YesAdditional CommentsAre the data and metadata consistent with relevant minimum information or reporting standards? See GigaDB checklists for examples <a href="http://gigadb.org/site/guide" target="_blank">http://gigadb.org/site/guide</a>YesAdditional CommentsIs the data acquisition clear, complete and methodologically sound?YesAdditional CommentsIs there sufficient detail in the methods and data-processing steps to allow reproduction?YesAdditional CommentsIs there sufficient data validation and statistical analyses of data quality? YesAdditional CommentsIt is a work that brings the geographical records of the genera involved in the transmission of Schistosomiasis in Uganda and the Democratic Republic of Congo. The specimens were identified at the genera level. No statistical analyses were performed. However, all the metadata generated was rigorously reviewed.Is the validation suitable for this type of data?YesAdditional CommentsIs there sufficient information for others to reuse this dataset or integrate it with other data?YesAdditional CommentsThese are important data, considering that very little has been studied on the African continent in relation to freshwater mollusks. Some important aspects that I believe should be considered and discussed: - The collected mollusks were identified at the genus level. Not all species of Biomphalaria, Bulinus and Radix are susceptible to Schistosoma mansoni. This needs to be clear in all parts of the manuscript and discussion needs to consider that. - I suggest that the authors indicate which species are transmitters and which occur in the countries studied. They can find this information from the literature, since they do not have this information based on their dataset. - On the map, I suggest that the boundaries of the provinces appear more clearly and put a map of the African continent indicating where each country studied is. The maps with the records can come as an enlargement from this map of the continent. - In conclusion, the authors mention that their results facilitate studies on the population structure of snails through size estimation. I remind the authors that when we talk about population, we are talking about the species level. It is not the case in the manuscript. - Also, it is important to keep in mind and to make clear in the manuscript that these molusks needs to be dissected to be identified. Snail images are not enough to identify the species. So, I suggest to modify this sentence making it clearer.Any Additional Overall Comments to the AuthorRecommendationMinor Revision